

# Impact of online learning on physical activity during COVID-19 lockdown period among female undergraduate students in Saudi Arabia: a cross-sectional study

Rania Almeheyawi[1], Alaa Alsini[1], Bayadir Aljadrawi[1], Layan Alshehri[1], Rawan Algethami[1], Razan Althobaiti[1], Ahlam Alrubeai[1], Hosam Alzahrani[1], Fahad Alshehri[1] and Yousef Alshehre[2]

[1] Department of Physical Therapy, College of Applied Medical Sciences, Taif University, Taif, Saudi Arabia
[2] Department of Physical Therapy, Faculty of Applied Medical Sciences, University of Tabuk, Tabuk, Saudi Arabia

## ABSTRACT

**Background.** During early 2020, because of the COVID-19 pandemic and related lockdown, most education systems—including universities—shifted from face-to-face classes to online learning. In Saudi Arabia, this might have contributed to a decreased level of physical activity (PA) and a concurrent increase in sedentary behaviour among young adults. This study aimed to investigate the impact of online learning on PA during the COVID-19 lockdown period among female undergraduate students in Saudi Arabia.

**Methods.** Data were collected through an online survey administered to participants. It consisted of three sections including demographic information, participants' perception towards online learning and PA, and PA level using the self-reported active-questionnaire survey tool. The association between online learning and PA was measured using linear regression. The statistical significance was set at $P < 0.05$.

**Results.** A total of 197 female undergraduate students were included; 95.4% of them were aged 18–24 years old, and 59.9% were in the normal body mass index range (18.5–24.9 kg/m$^2$). In terms of PA level, 55.3% were highly active, 33.5% were moderately active and 10.1% were low-active. In terms of students' perception of engaging in PA, 53.3% of students reported that engaging in PA definitely affected their psychological status. Moreover, compared with those attending <25 hours/week of online learning, those who attended >30 hours/week had lower PA ($r = -363.24$; 95% confidence interval (CI) $-593.97$, $-132.50$), followed by those attending 25–30 hours/week ($r = -277.66$; 95% CI $-484.65$, $-70.66$).

**Conclusion.** Online learning has negatively affected the PA level of female undergraduate students in Saudi Arabia during the COVID-19 lockdown period, in a dose-dependent manner. Moreover, this might affect their phycological status. Nevertheless, future studies are warranted to further investigate the relationship between PA level and psychological status.

Corresponding author
Rania Almeheyawi,
ralmeheyawi@tu.edu.sa

## INTRODUCTION

Physical activity (PA) is defined as any motion performed by the human body using the musculoskeletal system that causes energy expenditure (*Caspersen, Powell & Christenson, 1985*). Physical inactivity has been reported as a risk factor for multiple cardiovascular and chronic diseases, including hypertension, diabetes mellitus, cancer, osteoarthritis, obesity, and depression (*Warburton, Nicol & Bredin, 2006*). Modifying this risk factor has several benefits, as regularly performing PA can reduce the risk of premature death, disability, and chronic diseases (*Brownson, Boehmer & Luke, 2005*).

In the general population, engaging in higher levels of PA has been shown to be associated with better mental health (*Biddle, 2016*). Engaging in PA can reduce depression and anxiety (*Kandola et al., 2019*; *Rebar et al., 2015*) and increase self-esteem (*Robert, 2013*). Moreover, PA showed a generalized positive influence on mental health among undergraduate university students of both sexes (*Tyson et al., 2010*). A previous study conducted among middle-aged female participants reported that higher levels of PA were positively associated with higher self-esteem (*Dabrowska-Galas & Dabrowska, 2021*).

In early 2020, because of the COVID-19 pandemic, most education systems—including universities—shifted from on-campus classes (face-to-face learning) to online learning, allowing students to take their classes while socially isolating (*Zheng et al., 2020*). In Saudi Arabia, this transition continued until the end of the academic year 2020–2021. Online-based learning has several benefits, including time flexibility, time saving for both students and educators, and cost effectiveness related to traveling and space booking for on-campus learning activities. However, some challenges were faced and reported by students and educators during this online learning phase, including lack of student attention during online classes, lack of active student engagement with educators, lack of social interaction, difficulties in applying online learning to hands-on and practical courses, and most importantly, lack of physical movement during online learning (*Hiranrithikorn, 2019*; *Dung, 2020*; *Yuhanna, Alexander & Kachik, 2020*). These factors may contribute to an overall decreased level of PA and increased sedentary behaviour among young adults (*Zheng et al., 2020*). However, the impact of shifting to online-based learning on PA among undergraduate students in Saudi Arabia is still unknown.

Two studies investigated mental health among young adults between 18 and 25 years old (*Stroud & Gutman, 2021*) and anxiety and stress among university students (*Talapko et al., 2021*) during COVID-19 period, and both studies reported that female young adults between 18 and 25 years old were significantly more affected by COVID-19 on the mental health level and reported greater levels of stress and anxiety (*Stroud & Gutman, 2021*; *Talapko et al., 2021*). However, no consensus has been reached in literature regarding differences in PA between male and female university students during COVID-19 as one study reported similar reduction in PA in both male and female students (*Talapko*
*et al., 2021*), while another study revealed that 61% of female students reported that they performed PA less than usual compared to male students (56%) during COVID-19 (*Gallo et al., 2020*). Therefore, this study focused on female undergraduate students. It was hypothesized that online learning negatively affected PA levels during the COVID-19 lockdown period among female undergraduate students in Saudi Arabia. Therefore, the purpose of this study was to investigate the association between online learning and PA during the COVID-19 period among female undergraduate students in Saudi Arabia.

## MATERIALS & METHODS

### Study design

This cross-sectional study used an online survey questionnaire to collect data pertaining to PA from eligible participants. The data were collected between February and April 2021, when lockdown and online education were respectively reinforced and implemented in Saudi Arabia. This study was reported in accordance with the STROBE (Strengthening the Reporting of Observational Studies in Epidemiology) criteria. The protocol of this study was approved by the Institutional Review Board (IRB) of Taif University (IRB: 42-51).

### Participants

The estimated sample size was calculated based on the total number of female undergraduate students in Saudi Arabia ($n = 628,000$) (*Ministry of Education, 2021*) at a 95% confidence level and a 5% margin error. After calculation, the minimum estimated target sample size was 384 participants. The eligibility criteria for inclusion in this study were the following: female student, aged $\geq 18$ years, enrolled in an undergraduate university programme in Saudi Arabia, currently studying through online-based learning.

### Instruments

The data were collected using a self-constructed online questionnaire which was created and distributed using Google Forms. Eligible participants received a summary of relevant study details and a question about consent to participate in the study. The questionnaire was distributed using a non-random convenience sampling method by sending the questionnaire through emails or messages. The questionnaire consisted of three main sections. The first section included questions involving sociodemographic information, including age, height (in cm), weight (in kg), PA level, and average number of hours spent weekly on online learning during the COVID-19 pandemic period. The second section of the survey aimed to collect data related to individual perceptions of and personal experiences with online learning, PA, and psychological state. Two main direct questions were to be answered from multiple choices, ordered on a self-perception 5-point Likert scale (yes definitely, almost, sometimes, rarely, and no never). These questions were the following: "From your personal experience, did online learning affect your PA level?", and "From your personal experience, do you think that performing PA affected your psychological status?". The two questions have been developed by the researchers of this study and then have been piloted on a group of university students and reviewed by researchers before being included as a part of the study survey. The third section
included questions related to PA during the period of COVID-19, which were collected using the Active-Questionnaire (Active-Q) (*Bonn et al., 2012*). It is a reliable and valid self-reported questionnaire for measuring total energy expenditure and PA level among adults (18 years of age and older) and validated for adults between 18 and 45 years of age in large cohorts (*Bonn et al., 2012*). Active-Q is also suitable for web-based data collection, and it is user-friendly. It evaluates physical activity through queries in four distinct domains: (1) daily occupation, (2) transportation to daily occupation, (3) leisure time activities, and (4) sporting activities (*Bonn et al., 2012*). It calculates energy expenditure based on metabolic equivalent of task (MET) according to the nature of different activities and the number of minutes spent on each activity per week. It divides participants into four main categories: highly active (>1,000 MET-min/week), moderately active (500–1,000 MET-min/week), low-active (0–499 MET-min/week), and inactive (0 MET-min/week). Before starting this study, two professional translators unaware of the purpose of this study independently translated this questionnaire to Arabic language, then translated backward by two technical experts and agreement was reached between the translations. Moreover, the Arabic copy has been piloted on a group of university students before being used to collect data. Data were collected from participants anonymously as no data were collected related to personal information that might allow the researchers to recognize respondents such as name, identification number, or address.

## Data analysis

Data were collected using Google Forms and extracted using an Excel sheet. Body mass index (BMI) was calculated for all participants using self-reported weight (kg) and height (m). The SPSS statistical software package version 24 was used to analyse the data. Descriptive statistics as absolute and relative frequencies were reported as frequencies (numbers) and percentages. The association between online learning hours and PA was conducted using the general linear model, and the multiple regression was used to calculate the significance level for linear trend $p$ value. The models were adjusted for the BMI. The assumptions of the used parametric approaches were evaluated and met. The statistical significance was set at $P < 0.05$.

## RESULTS

### Participant characteristics

A total of 260 potential participants responded to the survey. However, 63 responses were excluded per the eligibility criteria (did not consent to participate, male participants, not students, younger than 18 years of age), leaving 197 eligible participants. All the 197 participants digitally signed a consent form in which the study goals and methods were outlined. Most participants were aged 18–24 years old (95.4%), and 59.9% were in the normal BMI range (BMI $=18.5–24.9$ kg/m$^2$).

Approximately half of the participants ($n = 93$, 47.2%) spent between 25 and 30 h per week on online learning, whereas 24.4% ($n = 48$) of them spent less than 25 h, and 28.4% ($n = 56$) spent more than 30 h per week on online learning. In terms of PA level, 55.3% ($n = 109$) of participants were classified as highly active, 33.5% ($n = 66$) moderately active,

**Table 1  Participants' sociodemographic data represented as frequency and relative frequency (%).**

| ($n = 197$) | Frequency (numbers) | Relative frequency (%) |
|---|---|---|
| **Age (years)** | | |
| 18–24 | 188 | 95.4 |
| <24 | 9 | 4.6 |
| **BMI (kg/m$^2$)** | | |
| Underweight (BMI >18.5) | 35 | 17.8 |
| Normal weight (BMI = 18.5–24.9) | 118 | 59.9 |
| Overweight (BMI = 25.0–29.9) | 35 | 17.8 |
| Obese (BMI ≥ 30) | 9 | 4.5 |
| **Online learning (hours/week)** | | |
| <25 | 48 | 24.4 |
| 25–30 | 93 | 47.2 |
| >30 | 56 | 28.4 |
| **PA level (MET-min/week)** | | |
| Inactive (0 MET-min/week) | 2 | 1.1 |
| Low (0–499 MET-min/week) | 20 | 10.1 |
| Moderate (500–1000 MET-min/week) | 66 | 33.5 |
| High (>1000 MET-min/week) | 109 | 55.3 |

Notes.
BMI, Body Mass Index; MET, Metabolic Equivalent; min, Minutes; PA, Physical Activity.

10.1% ($n = 20$) low-active, and 1.1% ($n = 2$) were inactive. The full sociodemographic data is presented in Table 1.

## Student perception of online learning and physical activity

Before collecting data related to PA and the average number of hours spent weekly on online learning, the participants were asked about their perception of online learning and PA. Table 2 shows the results expressed as frequency and relative frequency (%). When students were asked whether online learning affected their PA level in daily life, almost half of the participants ($n = 93$, 47.2%) answered "yes, definitely". Moreover, when asked if they thought that their psychological status was affected by their PA level, more than 53.3% responded "yes, definitely" ($n = 105$).

## Association between online learning and physical activity

Table 3 and Fig. 1 show the association between online learning and physical activity. The results showed a significant association between online learning and physical activity (Fig. 1). Compared with those attending <25 hours/week of online learning, those who attended >30 hours/week had lower PA ($r = -363.24$; 95% confidence interval (CI) [$-593.97, -132.50$]), followed by those attending 25–30 hours/week ($r = -277.66$; 95% CI [$-484.65, -70.66$]).

## DISCUSSION

Numerous studies have linked PA with academic performance (*Castelli et al., 2014*) and overall well-being (*Hyde, Maher & Elavsky, 2013*). It is well documented that PA is a

**Table 2  Participants' perception towards online learning and physical activity.**

| Participants' perception (n = 197) | Frequency (numbers) | Relative frequency (%) |
|---|---|---|
| **From your personal experience, did online learning affect your physical activity level?** | | |
| Yes, definitely | 93 | 47.2 |
| Almost | 51 | 25.9 |
| Sometimes | 26 | 13.2 |
| Rarely | 6 | 3.0 |
| No, never | 21 | 10.7 |
| **From your personal experience, do you think performing physical activity affected your psychological status?** | | |
| Yes, definitely | 105 | 53.3 |
| Almost | 40 | 20.3 |
| Sometimes | 22 | 11.2 |
| Rarely | 4 | 2.0 |
| No, never | 26 | 13.2 |

**Table 3  Association between online learning and physical activity.**

| Online learning, hours/week | Coefficient (95% CI)[a][b] | P value |
|---|---|---|
| <25 | Referent | – |
| 25–30 | −277.66 [−484.65, −70.66] | 0.002 |
| >30 | −363.24 [−593.97, −132.50] | 0.002 |

Notes.

CI, confidence interval.

[a]Adjusted for body mass index.

[b]Generalised linear model coefficients; coefficients indicate mean differences (in physical activity (metabolic-minutes/week)) between the reference category (<25 hours/week) and each of the other online learning groups, e.g., a value of −3 indicates that a specific category had a mean score that is 3 units lower than the referent group.

preventative measure for a variety of health problems, including obesity, hypertension, and diabetes (*Warburton, Nicol & Bredin, 2006*) and it can also reduce tension and anxiety levels (*Slimani et al., 2018*). The goal of the present study was to investigate the impact of online learning on the amount of PA that Saudi Arabian female undergraduate students engaged with during the COVID-19 lockdown. During the COVID-19 pandemic, online learning has been a pivotal tool for facilitating education when face-to-face learning was restricted. The results of the current study revealed that 43.6% of students were classified as having moderate or low-active PA. This finding is consistent with a previous study which found that Australian students showed a decline in PA during the lockdown (*Gallo et al., 2020*).

In 2021, a systematic review of 66 studies revealed that, except for individuals with eating disorders, the vast majority of research reported that PA levels dropped during the COVID-19 lockdown and most research indicated an increase in sedentary behaviour (*Stockwell et al., 2021*). The decline in PA is a major cause for worry owing to the potential

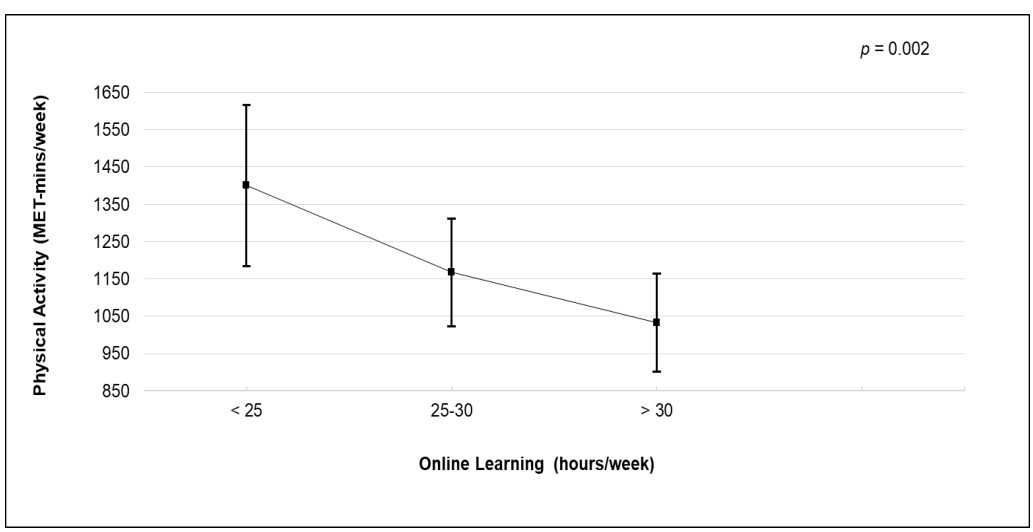

**Figure 1** Association between online learning (hours per week) and physical activity (minutes per week) using the general linear model.

repercussions to public health and the healthcare system in the long run. During the COVID-19 lockdown, numerous countries banned outdoor activities. However, outdoor exercising was possible in some countries, such as the United Kingdom, despite being under lockdown per their implemented social distancing guidelines (*Wickersham et al., 2021*). Interestingly, allowing outdoor exercising specifically among university students had a positive impact on their PA level, as reported by a study that investigated the changes in PA level among university students in the United Kingdom before and during the lockdown period. The study found that there was a gradual increase in steps walked per week following lockdown commencement. This increase was higher in males than in females (*Wickersham et al., 2021*).

The current study also showed that the time spent on online learning was inversely associated with the PA level. This is expected as many students attend classes from their residences, resulting in prolonged sedentary/screen-time periods. Moreover, the COVID-19 lockdown might have contributedto reduced PA frequency and duration, particularly for university students who previously used to walk or bike to work or class. Thus, it is logical to associate this finding with the loss of incidental walking during commute and daily activities such as walking between classes on campus (*Gallo et al., 2020*). Similar to our findings, a previous study reported that PA levels of female students fell dramatically due to the implementation of COVID-19-related isolation measures, despite calls for continued exercise during the pandemic (*Lippi, Henry & Sanchis-Gomar, 2020*).

Currently, 53.3% of female students in the present study perceived that their psychological status could be influenced by their PA level. Similar findings were reported by female university students in Spain, who showed higher levels of perceived anxiety than males (23) as female students perceived higher levels of apprehension, conscientiousness, neuroticism, and openness during the COVID-19 period as reported by *Rodriguez-Besteiro*

*et al. (2021)*; however, the study did not declare whether the data was collected during the application of lockdown restrictions in the country or not. Moreover, PA levels have been frequently associated with several mental health disorders (*Kandola et al., 2019*; *Rebar et al., 2015*), implying that declines in PA may result in an increase in negative mental health outcomes. Evidence has shown that during a COVID-19 lockdown, anxiety and despair levels rose significantly (*Schuch et al., 2020*).

Undoubtedly, shifting from face-to-face to online teaching and learning was challenging for most educators and students (*Almahasees, Mohsen & Amin, 2021*). Therefore, based on the results of the present study, diverse educational institutions may implement multidisciplinary interventions and awareness programs to counteract the negative impact of online learning on the students' PA level and improve their psychological status and mental health. This can be attained by encouraging students to proactively increase their PA levels while receiving online education as the impact of PA on mental health among undergraduate students has already been confirmed (*Rodríguez-Romo et al., 2022*). One of the limitations of this study is that while we employed a convenience sample strategy to readily reach a wide and varied group of participants from various universities in Saudi Arabia, this sampling method might have introduced some bias, potentially restricting the generalizability of the results. However, this was unavoidable to ensure the reliability of the findings as the relevant data needed to be collected in a relatively short period of time—while the COVID-19 lockdown restrictions in Saudi Arabia were in place. Another limitation is that the collected sample size was small which might increase the possibility for a type 2 error.

## CONCLUSIONS

This study hypothesized that online learning negatively affected the PA level of female undergraduate students in Saudi Arabia during the COVID-19 lockdown period. Indeed, online learning has negatively affected the PA level of female undergraduate students in Saudi Arabia during the COVID-19 lockdown period, as students who spent longer hours engaging in online education had lower PA levels. In turn, low PA levels might affect their psychological status; however, this relationship was not investigated in this study. Therefore, future studies are warranted to investigate this relationship.

## ACKNOWLEDGEMENTS

The authors would like to express their appreciation to all participants for their contributions to this work.

### Funding
The researchers received funding from the Deanship of Scientific Research, Taif University. The funders had no role in study design, data collection and analysis, decision to publish, or preparation of the manuscript.

## Grant Disclosures

The following grant information was disclosed by the authors:
Deanship of Scientific Research, Taif University.

## Competing Interests

The authors declare there are no competing interests.

## Author Contributions

- Rania Almeheyawi conceived and designed the experiments, analyzed the data, prepared figures and/or tables, authored or reviewed drafts of the article, and approved the final draft.
- Alaa Alsini conceived and designed the experiments, performed the experiments, prepared figures and/or tables, and approved the final draft.
- Bayadir Aljadrawi conceived and designed the experiments, performed the experiments, prepared figures and/or tables, and approved the final draft.
- Layan Alshehri conceived and designed the experiments, performed the experiments, prepared figures and/or tables, and approved the final draft.
- Rawan Algethami conceived and designed the experiments, performed the experiments, prepared figures and/or tables, and approved the final draft.
- Razan Althobaiti conceived and designed the experiments, performed the experiments, prepared figures and/or tables, and approved the final draft.
- Ahlam Alrubeai conceived and designed the experiments, performed the experiments, prepared figures and/or tables, and approved the final draft.
- Hosam Alzahrani analyzed the data, prepared figures and/or tables, authored or reviewed drafts of the article, and approved the final draft.
- Fahad Alshehri analyzed the data, authored or reviewed drafts of the article, and approved the final draft.
- Yousef Alshehre analyzed the data, authored or reviewed drafts of the article, and approved the final draft.

## Human Ethics

The following information was supplied relating to ethical approvals (*i.e.*, approving body and any reference numbers):

Institutional Review Board (IRB) at Taif University (IRB: 42-51)

## Data Availability

The raw data is available in the Supplemental File.

## Supplemental Information

Supplemental information for this article can be found online at http://dx.doi.org/10.7717/peerj.16579#supplemental-information.

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
