# Peer review of "Impact of online learning on physical activity during COVID-19 lockdown period among female undergraduate students in Saudi Arabia: a cross-sectional study"

_PeerJ, doi:10.7717/peerj.16579_

## Round 0.1 · original submission · Minor Revisions

Revise the manuscript according to the points raised by the reviewers.

·

Basic reporting

Clear and unambiguous, professional English used throughout., Literature references, sufficient field with relevant reference and within text citation. Figure easy to understand and sufficient, Professional article structure, figures, tables. Raw data shared. The hypothesis stated with relevant result and clear explanation.

Experimental design

The research is original and discuss an important topic. Research question well defined, relevant & meaningful. It is stated how research fills an identified knowledge gap. Rigorous investigation performed to a high technical & ethical standard. Method is sufficient in which used an ideal questionnaires.

Validity of the findings

The findings reflect the aim of the study and sounds great result with good analysis, discussion and conclusion is well stated.

Additional comments

its a good research and we need like this research in which reflect the Arabic community. the method is very good and the discussion is comprehensive.

Reviewer 2 ·

Basic reporting

The authors presented a well-written and structured research report and very succinct. The authors used appropriate literature and reference materials.

Line 47: this in-text reference (Caspersen C, Powell K, 1985) is not correct. I suggest you review the reference guide for this journal. I think your reference should be (Caspersen and Powell, 1985).

Introduction: I am not sure I saw any rationale or justification for the choice of only female undergraduate. Why not both male and female undergraduate, please provide a reason for this use of only female undergraduates.

Line 73-75: revise the statement to read “It was hypothesized that online learning negatively affected PA levels during the COVID-19 lockdown period among female undergraduate students in Saudi Arabia.

Figure 1 should be moved to after Table 3. I am not entirely sure that the style of Table 3 is correct. I presume you may want to check for published articles in this journal and format table 3 to the style of this journal. Another alternative would be to use the APA style for tables for your type of analysis.

Experimental design

Line 99: please specify the non-random sampling strategy used. Was this purposive, convenience or snowballing……

Line 104-110: were these two questions taken from a pre-existing questionnaire or were they self-developed? If the questions came from an existing questionnaire, can you indicate the validity.

Or, if these two questions were self-developed, did you conduct a face and content validity before using the two questions?

Lines 127-128: For data analysis, you did not report if your variables met the assumptions for linear regression and did not indicate if you adjusted for any potential confounding variables. I suggest that you indicate how your variables were used (i.e., continuous, ordinal, nominal etc) so that you can state the type of linear regression used for your data analysis. From Table 1, it appears that both variables were categorical variables – so I am not sure you used a linear regression. If you did, can you specify the exact type of linear regression that was used. I believe a chi-square analysis would also have been appropriate here as it appears your were looking at a bivariate relationship.

Validity of the findings

Lines 134-135: please change to “All the 197 participants digitally signed a consent form in which the study goals and methods were outlined.”
Line 137: Please remove this statement “The average weekly hours spent on online learning by the participants were investigated.” as it is redundant.
Lines168-169: The results of the current study revealed that 43.6% of students were classified as moderately or low-active. Please review this statement and consider changing to “The results of the current study revealed that 43.6% of students were classified as having moderate or low-active PA.”

Lines 177-179: Please add a reference at the end of this statement. “However, outdoor exercising was possible in some countries, such as the United Kingdom, despite being under lockdown per their implemented social distancing guidelines.” (add a reference here).
Line 194: this is a new sentence, so change “currently” to “Currently”
A potential limitation may also be the small sample size. I wondered if the sample size represents Saudi Arabia female undergraduates. I was just wondering if after the sample size calculation you did, you got 196, can this be correct??? As i think you should be getting 384 or thereabout…… http://www.raosoft.com/samplesize.html You can check with this sample size calculator. I think you did not consider response rate in your initial sample size calculations. I therefore suggest that you should indicate small sample size as a limitation of this study.
Line 223: change phycological status to ‘psychological status’

Additional comments

Generally very well written manuscript.

Reviewer 3 ·

Basic reporting

No comment.

Experimental design

Page 3 - material and methods

I would suggest that the authors make clear that it is an online survey on the study design.
Were the first questions developed by the authors? If yes, did they conduct any external validity exercises (e.g., consult with a panel of experts)? If they did not develop the questions, could the authors provide the references?
Have the authors pilot tested the questionnaire? Have they used a culturally validated version of Active Q? Were the completed questionnaires anonymized? If yes, what measures were undertaken to ensure that? Could the authors describe these methodological aspects on the manuscript?
Thank you.

Validity of the findings

No comment.

Additional comments

Congratulations on your manuscript.

---

## Round 0.2 · accepted · Accept

As the reviewer is satisfied with the modified manuscript, the manuscript is accepted for publication in its current form.

Reviewer 2 ·

Basic reporting

Clear and unambiguous, professional English used throughout.

Experimental design

Manuscript has met all necessary criteria stipulated by the journal

Validity of the findings

Important and interesting findings

Additional comments

All suggested comments added to by the authors. No further comments. Thank you and well done authors.